# The Effects of Textural Parameters of Zeolite and Silica Materials on the Protective and Functional Properties of Polymeric Nonwoven Composites

**Agnieszka Brochocka [1],\* , Aleksandra Nowak [1], Rafał Panek [2] and Wojciech Franus [2]**

1   Department of Personal Protective Equipment, Central Institute for Labour Protection—National Research Institute, 02-672 Warsaw, Poland; alnow@ciop.lodz.pl
2   Department of Geotechnical Engineering, Faculty of Civil Engineering and Architecture, Lublin University of Technology, 20-001 Lublin, Poland; r.panek@pollub.pl (R.P.); w.franus@pollub.pl (W.F.)
\*   Correspondence: agbro@ciop.lodz.pl; Tel.: +48-42-648-0225

**Abstract:** Zeolites are micro- and mesoporous aluminosilicate minerals (both natural and industrially produced) widely used as catalysts and sorbents in domestic and commercial water purification and separation technologies. Their ability to selectively adsorb gases (i.e., water vapor, carbon dioxide, and sulfur dioxide removal) from an air stream makes them suitable for applications in odor reducing media used in filtering facepiece respirators (FFRs). FFRs are multilayer products in which the most important role is played by high-performance melt-blown electret nonwovens modified with activated carbon to adsorb malodorous compounds. Replacing carbon sorbents with zeolites could increase the efficiency of odor abatement, thus alleviating work-related hazards for individuals exposed to malodorous substances with adverse effects on human well-being. The objective of the present work was to analyze the influence of the textural parameters of zeolite and mesoporous silica materials on the protective and functional properties of polymeric nonwoven composites containing them. In our experiments, the longest breakthrough time against ammonia vapor was found for a nonwoven composite containing the inorganic mesoporous silica material type MCM-41. It was also characterized by high filtration efficiency against aerosols with solid and liquid dispersed phases (97% and 99% for sodium chloride and paraffin oil mist, respectively) at an airflow resistance of approximately 330 Pa. In turn, the composites containing the molecular sieve (SM-zeolite ZSM-5) exhibited the longest breakthrough time for acetone and cyclohexane vapors at the maximum allowable concentrations of 235 ppm and 81 ppm, respectively. Basic filtration tests showed that the composite was 97% effective against both test aerosols at an airflow resistance of 283.5 Pa.

**Keywords:** zeolites; mesoporous silica; sorption capacity; textural parameters; polymer composites

## 1. Introduction

At a time of increasing public awareness, migration of urban dwellers to rural areas, spatial development of cities bringing residential areas closer to industrial facilities, and rapid economic growth the presence of noxious odors has often been reported to environmental law monitoring and enforcement agencies. Irrespective of their origin, malodorous compounds may lead to psychological discomfort or, in extreme cases, to pathological symptoms, including headaches, vomiting, diarrhea, upper airway and eye irritation, and even depression [1,2]. Due to the growing social awareness of the effects of air quality on human health and well-being, these issues have also become the subject of extensive research both in Poland and around the world [3–5]. Of particular note are studies on deodorization methods and evaluation tools measuring the performance of odor treatment systems

involving chemical (thermal and catalytic oxidation, ozonation), physical (condensation, adsorption, absorption), and biological (biofilters, bioscrubbers, other bioreactors) technologies [6–8]. However, the available literature mostly concerns industrial applications, with no data on the sorption of volatile compounds by the flat filtration/adsorption media used in respiratory protective devices.

Workplaces where the maximum allowable concentrations (MAC) of harmful substances are exceeded necessitate the use of gas filters and combined filters; otherwise, filtering facepiece respirators (FFRs) are worn to adsorb offensive odors. In addition to polymeric filtration layers, FFRs feature an activated carbon layer to reduce the amount of chemical compounds entering the breathing zone. In standard air-purifying respiratory protective equipment, air contaminated with volatile compounds flows through the sorption bed during inhalation, with the sorption capacity of that medium depending on the environmental conditions, airflow rate, and textural parameters of the material. Activated carbons used in adsorbers must meet a number of criteria, such as large adsorption capacity with respect to selected volatile compounds, low airflow resistance, low bulk density, large total micropore volume (to hold large amounts of adsorbed gases and vapors), and especially good adsorption kinetics and dynamics [9,10]. Standardized test conditions for investigating the adsorption properties of gas filters are specified in the standard EN 14387:2004+A1:2008. Studies on the effects of textural properties of activated carbons on their capacity to adsorb volatile chemicals have been widely reported in the Polish and international literature [11–14].

Due to the rising air pollution prevention requirements, researchers seek increasingly effective solutions that would be at once economical and ecologically safe. As a result, there has been a growing interest in zeolites-minerals which exhibit molecular-sieve, sorptive, and ion-exchange properties. Due to their widespread availability, low prices, and a suitable skeletal structure, they may be applied in broadly defined environmental technologies, from water and wastewater purification (removal of ammonium ions, radioactive elements, heavy metals, and petroleum hydrocarbon contaminants) to water and gas adsorption in agriculture and industry. While the international literature describes numerous zeolite applications in environmental engineering, and especially water and sewage treatment [15,16], there have been no reports concerning the use of such materials in FFRs. Furthermore, no sorption capacity evaluation methods or measurements have been developed for volatile compounds found in the workplace at sub-MAC concentrations. The replacement of carbon sorbents with microporous zeolites or mesoporous silica materials may improve the effectiveness of odor abatement, thus reducing the hazards associated with work under conditions of exposure to offensive odorants adversely affecting human well-being. The objective of the present study was to investigate the effects of textural parameters of zeolite and silica materials on the protective and functional properties of polymeric nonwoven composites containing them.

## 2. Materials and Methods

### 2.1. Zeolites and Silica Material

The studied composites were made with unmodified mesoporous silica materials and zeolites synthesized from fly ash by the classical hydrothermal method, as well as zeolite and silica materials modified with hexadecyltrimethylammonium (HDTMA) bromide supplied by the Lublin University of Technology. Data of the literature show the positive effect of surface modification of materials using surfactants from the group of quaternary ammonium salts (HDTMA type) on properties against volatile organic compounds (VOC) [17,18].

Na-A, Na-P1 zeolites were obtained by the hydrothermal conversion of fly ash with aqueous sodium hydroxide according to the following reaction:

$$flyash + x\,mol\cdot dm^{-3}\,NaOH \xrightarrow[temperature]{time} zeolite + residue \tag{1}$$

Zeolites were obtained on the prototype line of a technical quarter located at the Lublin University of Technology at the Faculty of Civil Engineering and Architecture. Using appropriate reaction parameters, such as the temperature, time, and substrate concentration, a zeolite containing zeolite phases was obtained: Na-A, Na-P1 [19]. The MCM-41 material was formed from the waste filtrate resulting from the synthesis of zeolites. The surfactant (hexadecyltrimethylammonium bromide (CTAB)) was the template for obtaining the structure of MCM-41 [20]. For the modifications obtained by synthesizing the materials, a HDTMA water solution in the amount of 24.4 mmol HDTMA/g material (NaP1, Na-A, MCM-41) was used. 200 cm$^3$ of distilled water was added to a portion of 200 g of material. The dissolved surfactant was poured into a beaker with modified material and stirred for 24 h at room temperature. The solution was then decanted, the material transferred to a dish, and dried at 60 °C [21].

The basic research method confirming the structure of materials obtained by synthesis was X-ray diffraction (XRD). The presence of individual crystalline phases for zeolites and mesoporous silica material was recognized on the basis of characteristic inter-plane distances [22,23]. Surface modification with HDTMA does not change the crystalline structure of materials [21].

The textural parameters of the studied zeolite and silica materials are shown in Table 1.

**Table 1.** Textural parameters of the studied zeolite and silica materials.

| Variant | BET Surface Area, m$^2$/g | BJH Pore Diameter, nm | BJH Average Pore Volume, cm$^3$/g |
|---|---|---|---|
| Zeolite Na-A | 8.52 | 5.11 | 0.013 |
| Zeolite Na-A HDTMA | 3.31 | 4.75 | 0.008 |
| Zeolite Na-P1 | 38.06 | 12.07 | 0.160 |
| Zeolite Na-P1 HDTMA | 5.03 | 3.63 | 0.006 |
| MCM-41 | 953.30 | 3.10 | 0.850 |
| MCM-41 HDTMA | 264.31 | 2.40 | 0.230 |
| Molecular sieve SM 4Å * | 3.40 | 4.50 | 0.006 |
| Molecular sieve SM ** | 325.60 | 2.21 | 0.130 |

** Molecular sieve, 4 Å = 0.4 nm, sodium aluminosilicate powder (Sigma-Aldrich Sp.zo.o., Poland). ** Molecular sieve in the form of powder-particles with a mean diameter of 3–5 μm (Sigma-Aldrich Sp. zo.o., Poland).

## 2.2. Filtering Nonvowen

Eight types of nonwoven composites were made using melt-blown technology. The matrix of the composites was Borealis HL 508J isotactic polypropylene (PP) supplied in the form of beads (NEXEO Solutions Poland Sp. zo.o., Poland), with the properties: Melting temperature (156–160 °C), melt flow index MFI (800 g/10 min), and density (50.3 g/cm$^2$).

In the melt-blown process, molten PP was air-blown to obtain elementary fibers of various thicknesses and lengths, with zeolite or silica materials introduced directly in the fiber stream. The resulting mixture of fibers and zeolite/silica material was electrostatically activated at 25 kV and settled on the collector to form a nonwoven zeolite/silica composite. The mean fiber diameter in the reference material was 968.78 nm.

The produced nonwoven composites are characterized in Table 2.

**Table 2.** Basic textural parameters of the produced nonwoven composites.

| Variant | BET Surface Area, m$^2$/g | Surface Density, g/m$^2$ | Composite Thickness, mm |
|---|---|---|---|
| PPQ + Na-A | 2.80 | 202.45 | 3.48 |
| PPQ + Na-A HDTMA | 2.32 | 214.01 | 2.48 |
| PPQ + Na-P1 | 13.38 | 340.57 | 3.70 |
| PPQ + Na-P1 HDTMA | 9.77 | 343.32 | 3.94 |
| PPQ + MCM-41 | 310.70 | 134.04 | 4.14 |
| PPQ + MCM-41 HDTMA | 6.27 | 299.94 | 4.49 |
| PPQ + SM 4 Å | 1.38 | 163.36 | 3.47 |
| PPQ + SM | 111.73 | 131.72 | 4.25 |

PPQ-polypropylene electrets melt-blown nonwoven.

BET surface area is one of the basic measures of adsorptive properties, especially in relation to adsorbents. The performed study allowed observations to be made that of all the variants, the high BET surface area values are characterized by: PPQ + MCM-41 (310.70 $g^2/m$) and PPQ + SM (111.73 $m^2/g$).

### 2.3. Surface Density

The surface mass of zeolite/silica composites expressed in grams per square meter was carried out in accordance with EN 965:1995 [24]. The surface mass, *Mp* was calculated according to the formula:

$$M_p = \frac{m * 10,000}{A} \tag{2}$$

where: *m*—mass of the sample in grams, *A*—area of the sample in $cm^2$

### 2.4. Composite Thickness

The thickness of composite material for sample dimensions of $13 \times 13$ cm was measured with the RAINBOW T model with a measuring range of 0-10 mm, the surface of the foot $(200 \pm 20)$ $mm^2$, and the pressure $(0.1 \pm 0.001)$ kPa according to the PN-EN ISO5084: 1999 standard [25].

### 2.5. Microscopic Evaluation of Morphological Properties

The morphological properties of zeolite and silica materials were studied using a Quanta 250 FEG scanning electron microscope (SEM, from FEI, Netherlands), while the nonwoven composites were examined using a NOVA NanoSem 230 high-resolution SEM (FEI, Netherlands). Samples were sputter-coated with carbon to improve conductivity and acquire better images. SEM studies were conducted under high vacuum (0.7 mbar) at an accelerating voltage of 30 to 60 kV, under $1000\times$ and $2000\times$ magnifications. Evaluation included the size distribution of fibers and zeolite/silica particles, including their homogeneity and attachment to the polymeric matrix. Fiber diameter and distribution were determined for a reference sample without zeolites or silica materials.

### 2.6. Textural Studies

Textural studies of zeolite and silica materials included determination of the most important surface and volume parameters using an ASAP 2020 specific area analyzer (Micromeritics Instrument Corporation) based on nitrogen vapor adsorption and desorption isotherms at liquid nitrogen temperature ($-194.85$ $^\circ$C). Measurements were conducted in the relative pressure, $p/p_0$, range from $1.5 \times 10^{-7}$ to 0.99.

The BET specific surface area of the tested nonwoven composites was studied using an AutosorbiQ analyzer (Quantachrome, USA), which measures sorption on the surface of solids by the static volumetric method. During the measurements, a set amount of adsorbate was introduced into a cell containing the adsorbent. The cell temperature was maintained at a constant level of $-196.15$ $^\circ$C. The samples were previously deaerated at 120 $^\circ$C for 12 h.

### 2.7. Studies of Protective and Functional Properties—Sorption Properties

The experimental stand shown in Figure 1 consists of six basic elements, that is, two *Dräger X-am 7000* gas analyzers, a pneumatic test chamber, an airflow control system equipped with *Red-y for gas flow* controllers, a humidifier, a recirculating cooler with a *HAAKE SC 100* thermostat, and an evaporator.

Gas analyzers were used to measure test gas concentrations upstream and downstream of the sample in the pneumatic holder. Vapors were generated using an evaporator in which the test substance was exposed to compressed air. Airflow controllers were used to obtain a desired concentration of the test substance (during the measurements, gas concentration should not exceed $\pm$ 5 ppm of the initial value) at a relative humidity of $(70 \pm 5)$% and a temperature of $(21 \pm 1)$ $^\circ$C [26]. The concentrations of the test substances, acetone (235 ppm), cyclohexane (81 ppm), and ammonia (18.7 ppm), corresponded

to their MAC levels pursuant to the Polish Regulation of the Minister of Labor and Social Policy of 12 June 2018 [27]; the volumetric flow rate was 30 L/min.

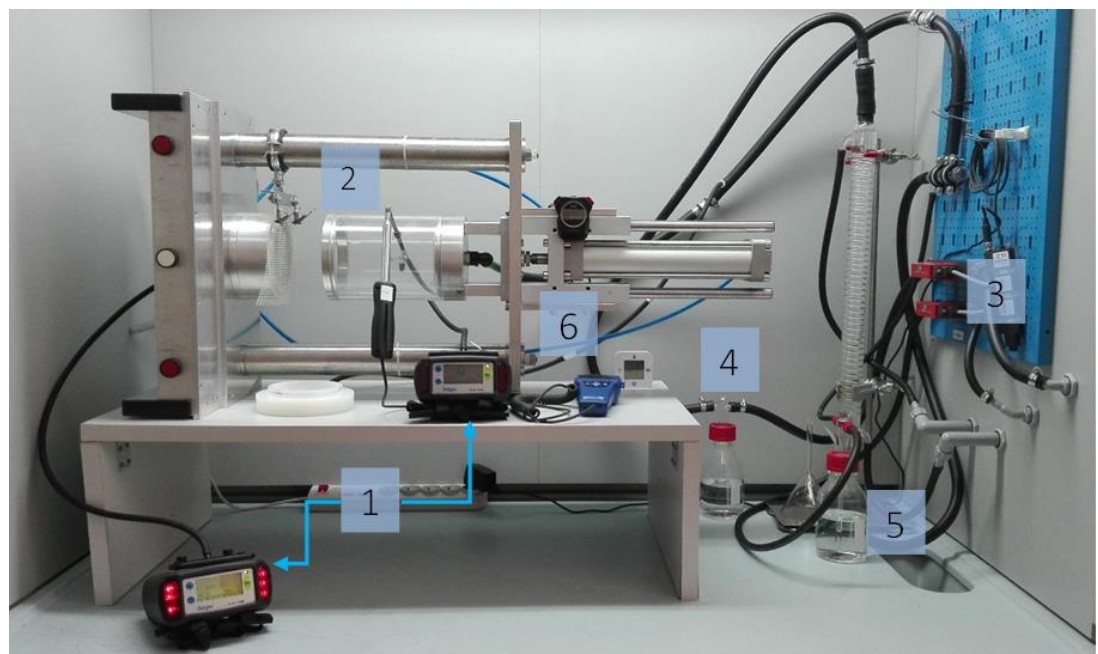

**Figure 1.** Experimental stand for measuring the sorption capacity of volatile compounds for flat textile products: 1—gas analyzers, 2—pneumatic test chamber, 3—airflow control system, 4—humidifier, 5—evaporator, 6—thermohygrometer.

Breakthrough time tests for the studied zeolite and silica composites were conducted for three substances: Acetone, cyclohexane, and ammonia.

On the basis of the performance tests, the sorption capacity of the tested variants with the addition of zeolites and siliceous material was calculated from the following equation:

$$C = \frac{V \cdot C_{in} \cdot tb_{100\ (corr)} \cdot 10^{-6}}{m_a} \tag{3}$$

where: $V$—volumetric flow rate of the test mixture [L/min], $C_{in}$—test concentration [mg/m$^3$], $tb_{100\ (corr)}$—corrected breakthrough time of the deposit [min], $m_a$—adsorbent mass [g].

In order to properly assess the performance of the adsorptive materials used, the partition coefficient was determined in accordance with the currently available literature [28,29]. Thus, it can be obtained from the following equation:

$$PC = \frac{Capacity}{P \times molecular\ weight} \tag{4}$$

where: *Capacity*—the adsorption capacity (mg/g); *P*—partial pressure (Pa) of the ammonia/acetone/cyclohexane gas at the saturation.

### 2.8. Studies of Protective and Functional Properties—Penetration by Sodium Chloride Aerosol

The filtration performance of the studied nonwoven composites with respect to solid aerosol particles was evaluated using an apparatus for measuring penetration by sodium chloride (NaCl) aerosol with a solid dispersed phase pursuant to the European standards concerning FFR requirements and testing [30,31]. According to the adopted methodology, the study measured changes in filtration efficiency in the third minute of experiment (during the initial filtration stage). The apparatus consisted

of an installation supplying compressed air, which was purified and dried by a system of filters. The conditioned air was directed to an atomizer containing an aqueous NaCl solution. The generated polydisperse aerosol was directed to the chamber with the pneumatic sample holder in which the tested nonwoven was installed. A differential pressure gauge was connected to the inlet and outlet of the chamber to record airflow resistance. A tube was placed downstream of the sample holder to collect air samples for analyzing the concentration of the test aerosol. The concentration of the test aerosol at the inlet was $(8 \pm 4)$ mg/m$^3$ and was controlled using an integrated flame photometer. The photometer measured the intensity of the light beam emitted by a hydrogen burner as an indicator of the aerosol concentration, with a measurement range of 0.001–100%.

### 2.9. Studies of Protective and Functional Properties-Penetration by Paraffin Oil Mist

Paraffin oil mist penetration is a standard test method for assessing the filtration performance of nonwovens against liquid aerosols. Measurements were conducted pursuant to the methodology described in the European standards concerning FFR requirements and tests [30,31]. The aerosol produced in a Lorentz AGW-F/BIA generator was passed at flow rate of 95 L/min through nonwoven samples placed in an FH 143/149 pneumatic holder with a diameter of 100 mm. The size distribution of paraffin oil mist particles was log-normal with a median Stokes diameter of 0.4 μm. Aerosol concentrations upstream and downstream of the tested nonwoven samples were measured using a Lorenz AP2E laser photometer.

The paraffin oil mist penetration ($P_{POM}$) coefficient was calculated from the following formula:

$$P_{POM} = \frac{l_2 - l_0}{l_1 - l_0} \cdot 100\% \tag{5}$$

where: $l_0$, $l_1$, and $l_2$ designate photometric measurements for pure air and air with oil mist upstream and downstream of the nonwoven sample, respectively.

The results were read after three minutes of the experiment, which means that the coefficient was determined during the initial filtration phase.

### 2.10. Studies of Protective and Functional Properties—Airflow Resistance

Airflow resistance was determined pursuant to the European standards, EN 149:2001+A1:2009 and EN 13274-3:2008, concerning respiratory protective devices [30,32]. Air was passed through the nonwoven samples at a constant volumetric flow and the downstream pressure differential was measured with respect to the atmospheric pressure. Tests were conducted at a flow rate of 95 L/min, which corresponds to respiratory minute ventilation during strenuous work. Pressure was read from a CMR-10 A digital differential micromanometer.

## 3. Results

### 3.1. Microscopic Evaluation of Morphological Properties

The morphological properties of the studied zeolite and silicon materials as well as the nonwoven composites are shown in Figures 2–6.

Figure 2a shows zeolite Na-A produced by the hydrothermal method from the dissolved crystalline phase of fly ash. Zeolite Na-A particle represents the Linde A-like framework (code LTA according to IZA atlas structure) in which 6-membered rings form regular 4.1 × 4.1 Å channels. Its 2–5 μm crystals have a regular structure. In the case of the Na-A composite (Figure 2b), individual zeolite particles were attached to elementary polymer fibers with a uniform distribution across the nonwoven. Zeolite Na-A impregnated with HDTMA is shown in Figure 2c; its crystals are covered with a thin layer of HDTMA. Zeolite Na-A HDTMA particles incorporated in the nonwoven composite (Figure 2d) had a tendency to agglomerate. In addition to individual particles attached to the fibers, large quantities of agglomerated Na-A HDTMA powder were found mostly in spaces between the

entangled polymeric fibers. Zeolite modification led to agglomeration already at the stage of zeolite introduction in the melt-blown process. Approximately 55 g of zeolites Na-A and Na-A HDTMA were incorporated per nonwoven sheet.

Figure 3a presents the morphology of Zeolite Na-P1 formed on ash grains as a result of a hydrothermal reaction between an aqueous solution of sodium hydroxide and ash components, such as: Aluminosilicate glass, quartz, and mullite. Zeolite Na-P1 represents a gismondine-like framework (code GIS according to the IZA atlas structure) in which two 4-membered rings form an 8-membered channel measuring 3.1 × 4.5 Å and 2.8 × 4.8 Å along their axes, respectively. Zeolite Na-P1 plates-like crystals with sharp edges form characteristic 1–3 μm rosettes. In turn, the modified zeolite Na-P1 HDTMA exhibits rounded crystal edges (Figure 3c). Approximately 123 g of zeolites Na-P1 and Na-P1 HDTMA were incorporated per nonwoven sheet. Both zeolite types had a tendency to agglomerate, as shown in Figure 3b,d.

The spaces between fibers are filled with a large number of agglomerates, which are much larger than the polymeric fiber diameters.

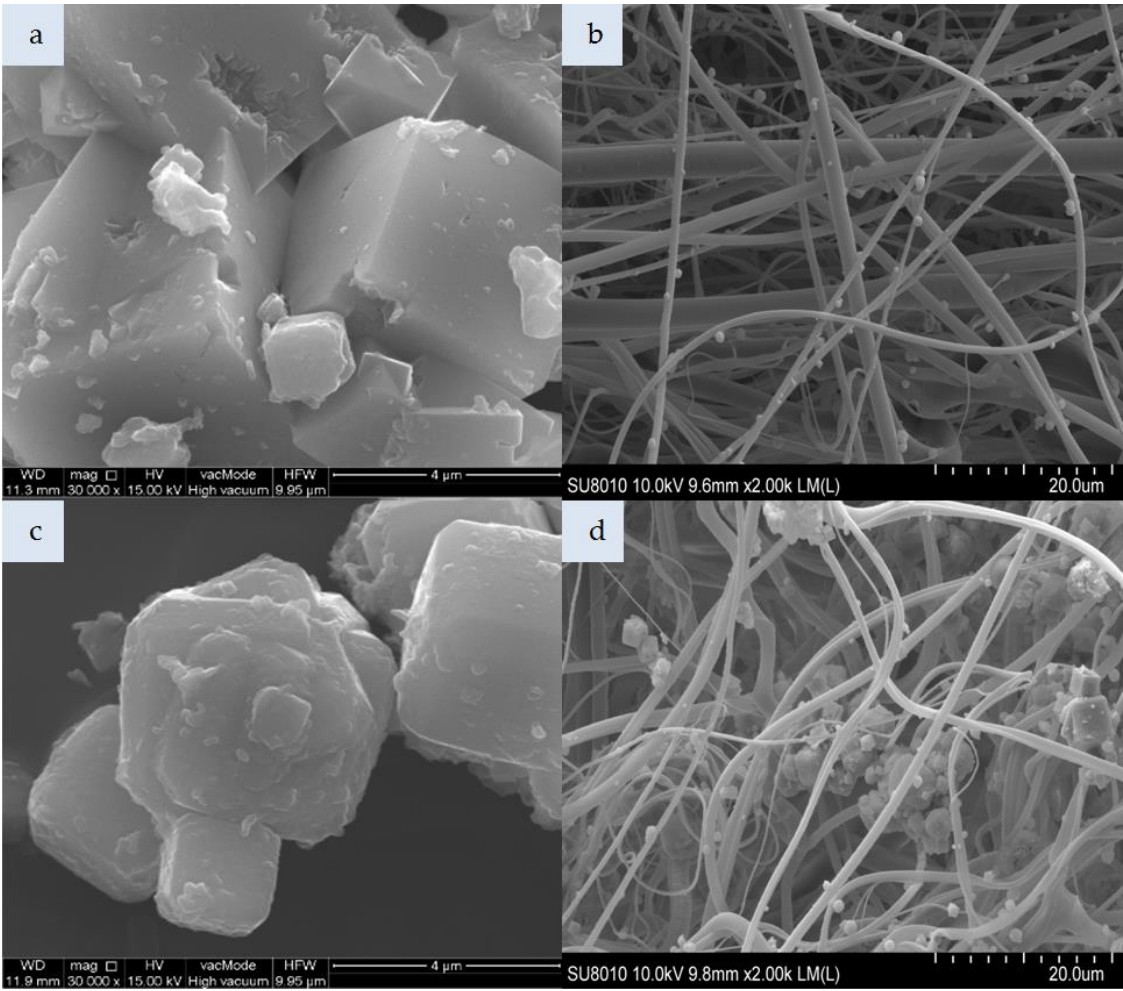

**Figure 2.** SEM photomicrograph: (**a**) Unmodified zeolite Na-A, (**b**) nonwoven composite with Na-A, (**c**)zeolite Na-A HDTMA, (**d**)nonwoven composite with Na-A HDTMA (magnification: a,c—x30.00k; b,d—x2.00k).

Figure 4a presents cylindrical forms of MCM-41 with a hexagonal structure. That material consists of 99% silicon, while the impregnated variant additionally contains HDTMA (Figure 4c). The incorporation of MCM-41, which has a low specific weight, in the polymeric nonwoven structure resulted in its uniform distribution (Figure 4b). Both individual particles and small agglomerates of

this silica material were attached to the surface of polymeric fibers. The HDTMA modification of MCM-41 led to a tendency to form large agglomerates locked in the nonwoven scaffold (Figure 4d). The SM 4Å molecular sieve particles shown in Figure 5a have a more complex form as compared to SM particles (Figure 6a). The SM molecular sieve is zeolite ZSM-5, which consists of a system of pores formed by straight, parallel 10-membered channels measuring 5.3 × 5.6 Å as well as sinusoidal 10-ring channels (5.5 × 5.1 Å). The SM molecular sieve, which was introduced into the polymeric nonwoven in small quantities (approximately 18 g per sheet), is uniformly distributed (Figure 6b). Individual particles were attached to elementary fibers, with some fine agglomerates in interfiber spaces. Interestingly, more agglomerates were found attached to the surface of polymeric fibers. In turn, the SM 4Å molecular sieve had a strong tendency to form agglomerates (much larger than elementary fiber diameters) in interfiber spaces (Figure 5b).

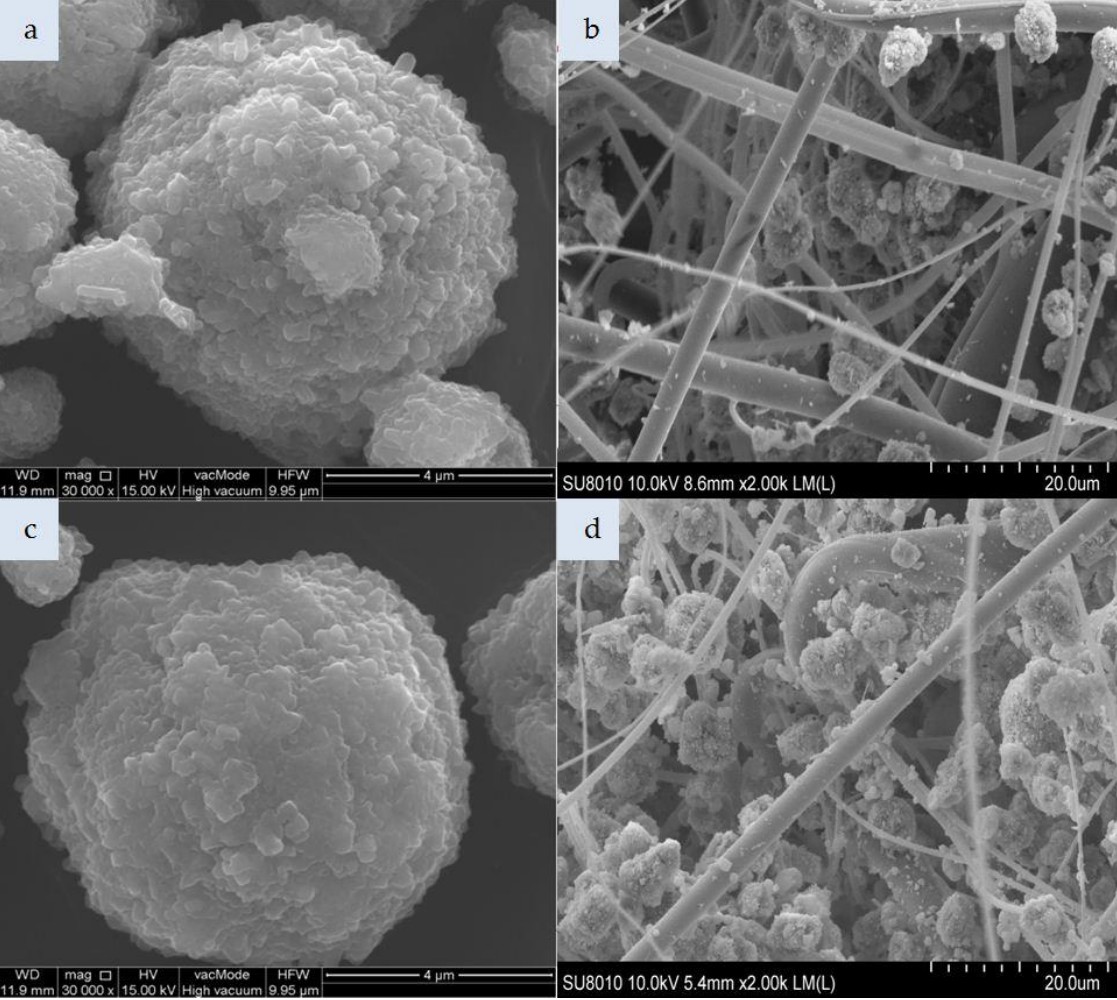

**Figure 3.** Photomicrograph: (**a**) unmodified zeolite Na-P1, (**b**) nonwoven composite with zeolite Na-P1, (**c**) zeolite Na-P1 modified with HDTMA, (**d**) nonwoven composite with zeolite Na-P1 HDTMA (magnification: a,c—x30.00k; b,d—x2.00k).

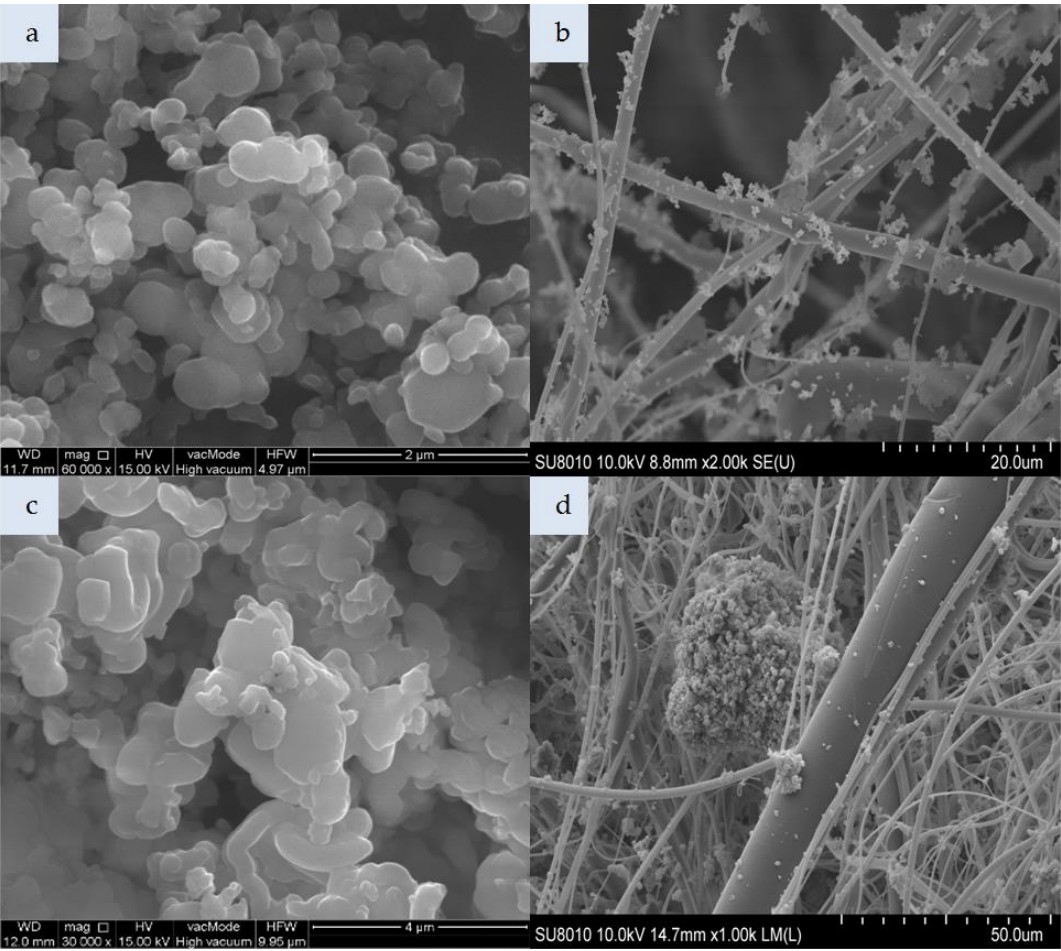

**Figure 4.** SEM photomicrograph: (**a**) Mesoporous silica material (MCM-41), (**b**) nonwoven composite with MCM-41, (**c**) MCM-41 modified with HDTMA, (**d**) nonwoven composite with MCM-41 HDTMA (magnification: a—x60.00k, b—x2.00k c—x30.00k; d—x1.00k).

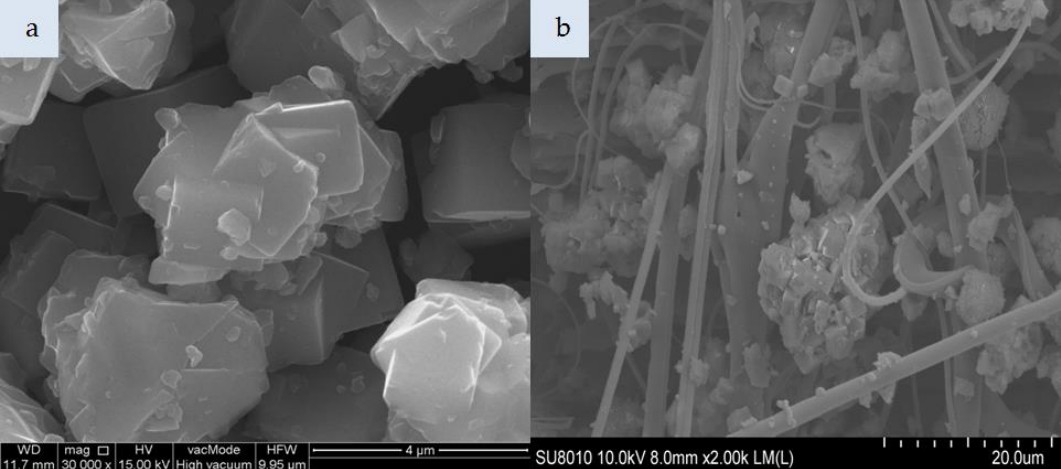

**Figure 5.** SEM photomicrograph: (**a**) Molecular sieve SM 4Å, (**b**) nonwoven composite with SM 4Å (magnification: a—30.00k, b—2.00k).

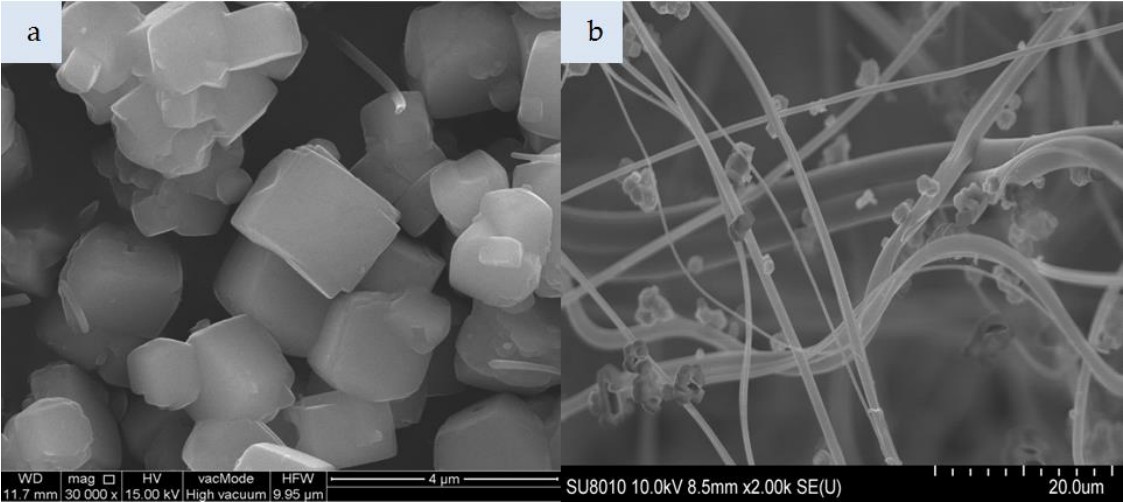

**Figure 6.** SEM photomicrograph: (**a**) Molecular sieve SM, (**b**) nonwoven composite with SM (magnification: a—30.00k, b—2.00k).

### 3.2. Sorption Properties

Breakthrough time results for the developed variants of nonwoven zeolite and silica composites challenged with ammonia, cyclohexane, and acetone are presented in Figures 7–9, respectively.

The obtained results show that the developed variants of nonwoven zeolite and silica composites selectively adsorb the vapors of volatile compounds.

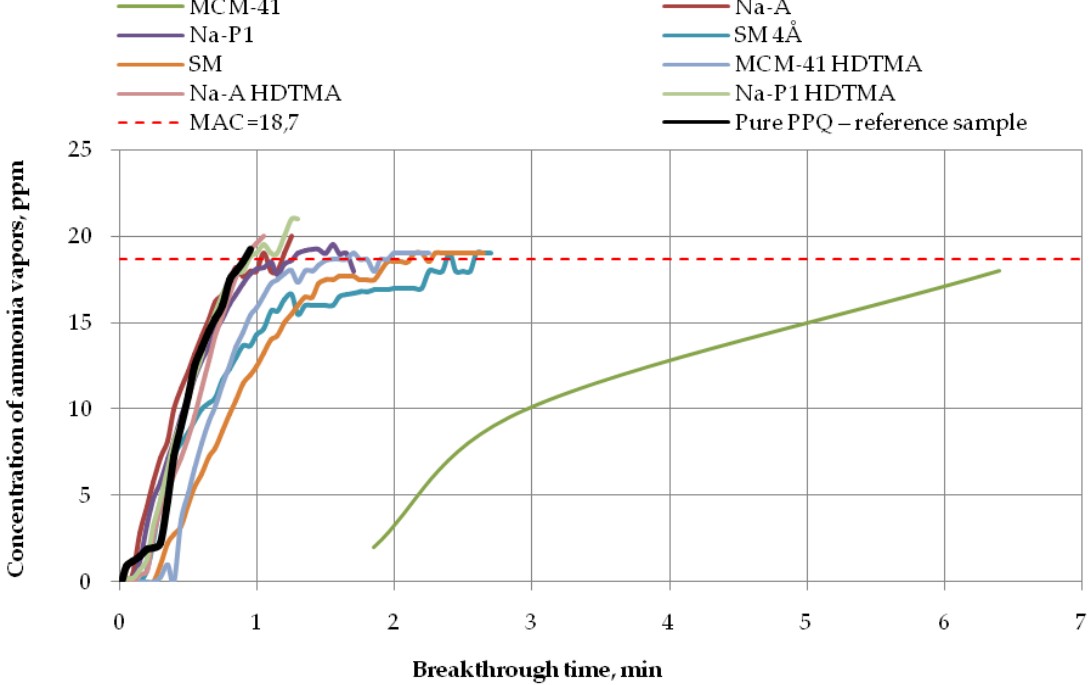

**Figure 7.** Breakthrough time for nonwoven zeolite and silica composites challenged with ammonia at MAC = 18.7 ppm.

The longest breakthrough time against ammonia vapors (more than 6 min) at 18.7 ppm was found for the nonwoven composite containing the mesoporous silica material PPQ + MCM-41, while the other composites incorporating zeolite, whether modified or not, practically did not exhibit protective properties against this compound (Figure 7). These results are in accordance with the literature data [33] as interactions occurring between modified zeolites and ammonia effectively improve their

ionic conductivity. In addition, elevated temperature accelerates ammonia adsorption/desorption kinetics, thus decreasing the adsorption capacity of zeolites. The reaction time was 1 min for the composite containing mesoporous silica material modified with HDTMA, 2 min for the composite with the molecular sieve SM, and more than 2 min for the one with the molecular sieve SM 4Å, in relation to the reference sample. The longest breakthrough time against ammonia vapors was obtained for the mesoporous silica material characterized by a large specific surface area (953.30 m$^2$/g) and the greatest pore volume among the tested materials (0.85 cm$^3$/g). These results are consistent with other reports [34–36]. Materials characterized by very high specific surface areas, containing pores of various shapes and sizes throughout their volume, ensure better contact between the active sites on their surface with adsorbate particles. There is a close relationship between the size of the specific area and the pore size.

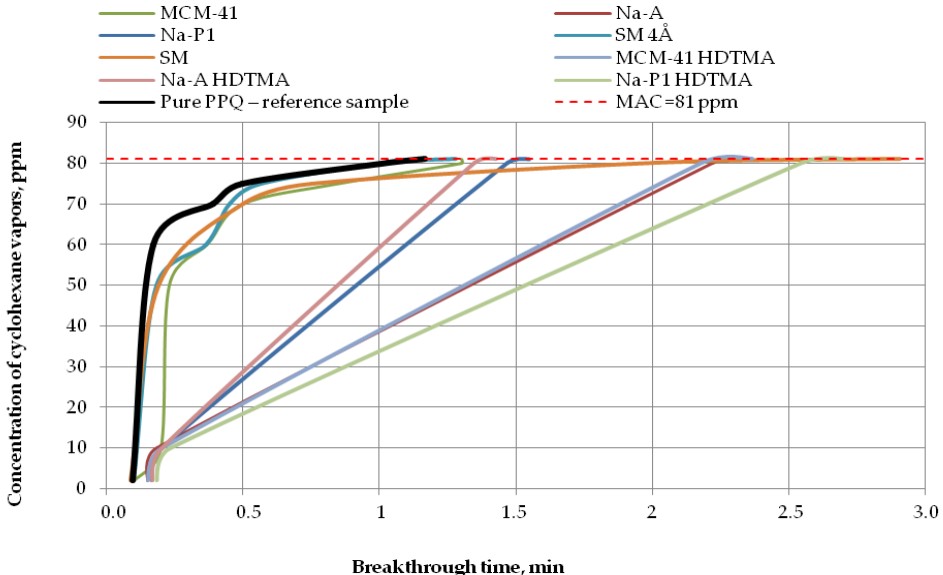

**Figure 8.** Breakthrough time for nonwoven zeolite and silica composites challenged with cyclohexane at MAC = 81 ppm.

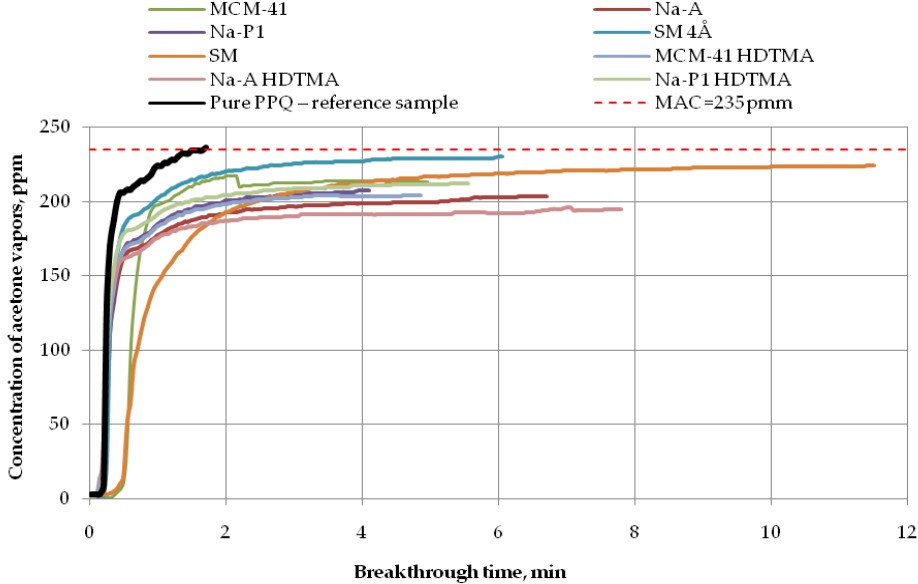

**Figure 9.** Breakthrough time for nonwoven zeolite and silica composites challenged with acetone at MAC = 235 ppm.

In the case of cyclohexane vapors at a concentration of 81 ppm, the nonwoven composite containing zeolite Na-A HDTMA revealed the maximum penetration of the test substance after 1 min, the composite containing zeolite Na-P1 after 1.5 min, and composites with MCM-41 HDTMA, Na-A, and the molecular sieve SM after 2 min. The longest reaction time to cyclohexane vapors, more than 2.5 min, was found for the composite containing zeolite Na-P1 HDTMA (Figure 8). Thus, zeolite modification led to a longer breakthrough time against cyclohexane.

Finally, the longest breakthrough time against acetone vapors (more than 11 min) at 235 ppm was found for the nonwoven composite containing the molecular sieve SM (zeolite ZSM-5), as shown in Figure 9. The application of the molecular sieve SM 4Å resulted in 232 ppm downstream of the sample after 6 min. In turn, after 5 min, that concentration ranged from 200 ppm to 220 ppm for the zeolite composite PPQ + Na-P1 HDTMA and the silica composites PPQ + MCM-41 and PPQ + MCM-41 HDTMA. The composite with zeolite Na-A exhibited a downstream concentration of acetone vapors of 200 ppm after 6 min. This means that the composite with the longest breakthrough time contained the molecular sieve SM, characterized by a specific surface area of 352.6 m$^2$/g at a pore volume of 0.13 cm$^3$/g. The following (Table 3) is a presented summary of data obtained for the assessment of the action of adsorbents used to remove ammonia, acetone, and cyclohexane from the air.

**Table 3.** Summary of data obtained for the assessment of the action of adsorbents used to remove ammonia, acetone, and cyclohexane from the air.

| Type of Tested Gas | Type of Sorbent | Sorbent Code | Concetration, ppm | Partial Pressure, Pa | Capacity (C), mg g$^{-1}$ | Partition Coefficient (PC), mol kg$^{-1}$ Pa$^{-1}$ |
|---|---|---|---|---|---|---|
| AMMONIA | Zeolite | Na-A | 18.7 | 1.87 | 0.0003 | $8.77 \times 10^{-6}$ |
| | | Na-P1 | | 1.87 | 0.0002 | $5.21 \times 10^{-6}$ |
| | | SM 4A | | 1.87 | 0.0008 | $2.48 \times 10^{-5}$ |
| | | SM | | 1.87 | 0.0013 | $4.17 \times 10^{-5}$ |
| | | Na-A H | | 1.87 | 0.0002 | $5.84 \times 10^{-6}$ |
| | | Na-P1 H | | 1.87 | 0.0001 | $3.01 \times 10^{-6}$ |
| | MSM * | MCM-41 | | 1.87 | 0.0018 | $5.76 \times 10^{-5}$ |
| | | MCM-41 H | | 1.87 | 0.0002 | $5.41 \times 10^{-6}$ |
| ACETONE | Zeolite | Na-A | 235 | 23.5 | 0.7267 | $5.32 \times 10^{-4}$ |
| | | Na-P1 | | 23.5 | 0.2003 | $1.47 \times 10^{-4}$ |
| | | SM 4A | | 23.5 | 1.1666 | $8.55 \times 10^{-4}$ |
| | | SM | | 23.5 | 4.0716 | $2.98 \times 10^{-3}$ |
| | | Na-A H | | 23.5 | 0.7321 | $5.36 \times 10^{-4}$ |
| | | Na-P1 H | | 23.5 | 0.2708 | $1.98 \times 10^{-4}$ |
| | MSM * | MCM-41 | | 23.5 | 1.5640 | $1.15 \times 10^{-4}$ |
| | | MCM-41 H | | 23.5 | 0.2757 | $2.02 \times 10^{-4}$ |
| CYKLOHEXANE | Zeolite | Na-A | 81 | 8.1 | 0.0501 | $7.34 \times 10^{-5}$ |
| | | Na-P1 | | 8.1 | 0.0146 | $2.14 \times 10^{-5}$ |
| | | SM 4A | | 8.1 | 0.0434 | $6.37 \times 10^{-5}$ |
| | | SM | | 8.1 | 0.1860 | $2.73 \times 10^{-4}$ |
| | | Na-A H | | 8.1 | 0.0279 | $4.09 \times 10^{-5}$ |
| | | Na-P1 H | | 8.1 | 0.0251 | $3.69 \times 10^{-5}$ |
| | MSM * | MCM-41 | | 8.1 | 0.0783 | $1.15 \times 10^{-4}$ |
| | | MCM-41 H | | 8.1 | 0.0267 | $3.92 \times 10^{-5}$ |

* Microporous silica material.

The results presented (Table 3) showed that the highest values of sorption capacity (C = 0.0018 mg/g) and PC against ammonia are characterized by MCM-41 (PC = $5.76 \times 10^{-5}$ mol kg$^{-1}$ Pa$^{-1}$), whose BET surface area is the highest (310.70 g/m$^2$) in comparison with all sorbents used. In the case of VOCs, i.e., acetone and cyclohexane, the highest PC indicator is characterized by the zeolite material SM, for which the PC value is $2.98 \times 10^{-3}$ mol kg$^{-1}$ Pa$^{-1}$ and $2.73 \times 10^{-4}$ mol kg$^{-1}$ Pa$^{-1}$, respectively. For capacity, these values were: Acetone—4.0716 mg/g; cyclohexane—0.1860 mg/g), respectively. It is worth emphasizing that this material showed the second highest BET surface area (111.73 g/m$^2$) in relation to all adsorbents used in the above work. Based on the obtained data, it can be concluded that it has the best adsorption properties to VOCs compared to the rest of the modifiers. It was noted that the large structure of pores and the BET surface area of the adsorbents used have a large impact on the breakthrough time, capacity, and partition coefficient when saturated with test chemical substance vapors. Due to the small size of the specific surface

area and pore volume, the remaining group of modifiers used did not give the highest ammonia and VOCs adsorption capacity.

### 3.3. Penetration by NaCl Aerosol

The mean NaCl aerosol penetration results for the studied nonwoven composites containing zeolite and silicate materials are given in Table 4, with relative changes shown in Figure 10. Deterioration of protective and functional properties of the composite materials in relation to the reference sample is marked with negative values, while the improvement of protective and functional with positive values. On the basis of the statistical analysis performed, no significant differences were found, although absorbent materials with different textural parameters were used.

**Table 4.** Mean NaCl aerosol penetration for nonwoven composites containing zeolite and silica materials (descriptive statistics).

| No. | Variant | Mean NaCl Aerosol Penetration, % | Standard Deviation | Median | MAX, % | MIN, % |
|-----|---------|--------------------------------|--------------------|--------|--------|--------|
| 1 | PPQ + MCM-41 | 2.59 [a] | 0.70 | 2.34 | 3.94 | 1.89 |
| 2 | PPQ +Na-A | 2.51 [a] | 1.18 | 2.05 | 4.40 | 1.32 |
| 3 | PPQ + Na-P1 | 4.27 [a] | 1.52 | 5.12 | 6.08 | 2.01 |
| 4 | PPQ + SM 4Å | 2.91 [a] | 1.12 | 2.49 | 5.40 | 2.17 |
| 5 | PPQ + SM | 2.58 [a] | 0.34 | 2.47 | 3.31 | 2.26 |
| 6 | PPQ + MCM-41 HDTMA | 1.94 [a] | 0.36 | 1.95 | 2.49 | 1.45 |
| 7 | PPQ + Na-A HDTMA | 16.25 [b] | 7.96 | 18.30 | 24.65 | 3.74 |
| 8 | PPQ + Na-P1 HDTMA | 3.33 [a] | 1.62 | 2.62 | 6.92 | 2.29 |
| 9 | Pure PPQ reference sample | 2.47 | 0.26 | 2.36 | 2.93 | 2.21 |

[a,b]—statistically significant differences were found for the mean values marked with different letters (ANOVA, $\alpha = 0.05$; Tukey's test, $\alpha = 0.05$).

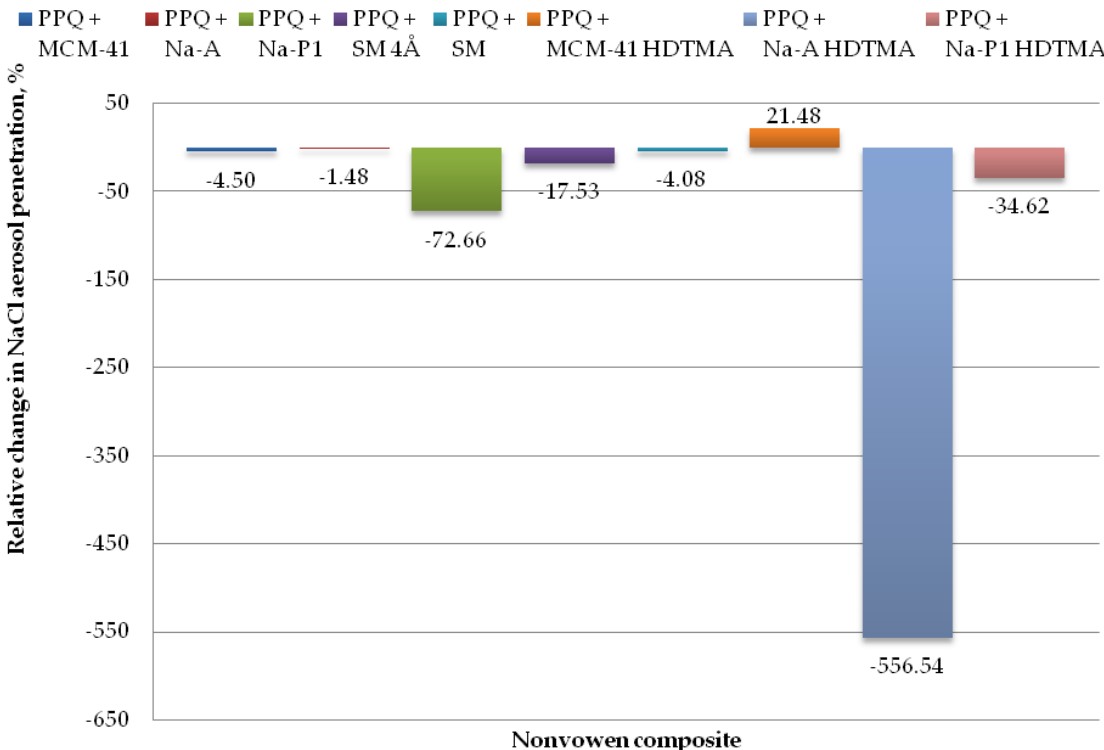

**Figure 10.** Relative change in NaCl aerosol penetration for nonwoven zeolite and silica composite.

The lowest values for penetration by NaCl aerosol were obtained for the composite containing the zeolite material Na-A, the unmodified and modified silica materials MCM-41 and MCM-41 HDTMA, and the molecular sieve SM, at 2.51, 2.59, 1.94, and 2.58%. Changes in NaCl aerosol penetration for those composite variants relative to the reference sample amounted to 1.48, 4.50, 21.48,

and 4.08%. The introduction of mineral additives in the structure of elementary fibers in the melt-blown process improved filtration efficiency as compared to the reference nonwoven only in one case, that is, PPQ + MCM-41 HDTMA.

The highest NaCl penetration was found for the nonwoven composite containing the modified zeolite Na-A HDTMA, at 16.25%, with the relative penetration change being 556.54%. This is attributable to the fact that the applied zeolite contains chemical elements that may neutralize the electrostatic charges induced during the technological process.

### 3.4. Penetration by Paraffin Oil Mist

The mean paraffin oil mist aerosol penetration results for the tested nonwoven composites containing zeolite and silica materials are given in Table 5, with relative changes shown in Figure 11. Deterioration of protective and functional properties of the composite materials in relation to the reference sample is marked with negative values, while the improvement of protective and functional with positive values. Based on the statistical analysis performed, no significant differences were found, although absorbent materials with different textural parameters were used.

**Table 5.** Mean paraffin oil mist aerosol penetration for nonwoven composites containing zeolite and silica materials (descriptive statistics).

| No. | Variant | Mean Paraffin Oil Mist Aerosol Penetration, % | Standard Deviation | Median | MAX, % | MIN, % |
|---|---|---|---|---|---|---|
| 1 | PPQ + MCM-41 | 1.30 [a] | 0.27 | 1.20 | 1.80 | 0.98 |
| 2 | PPQ + Na-A | 0.69 [a] | 0.36 | 0.80 | 1.20 | 0.59 |
| 3 | PPQ + Na-P1 | 2.19 [a] | 0.58 | 2.20 | 3.10 | 1.10 |
| 4 | PPQ + SM 4Å | 3.18 [a] | 0.19 | 3.20 | 3.40 | 2.90 |
| 5 | PPQ + SM | 2.73 [a] | 0.91 | 2.75 | 4.00 | 1.50 |
| 6 | PPQ + MCM-41 HDTMA | 0.51 [a] | 0.04 | 0.49 | 0.58 | 0.48 |
| 7 | PPQ + Na-A HDTMA | 14.25 [b] | 0.83 | 14.50 | 15.00 | 13.00 |
| 8 | PPQ + Na-P1 HDTMA | 2.12 [a] | 0.91 | 1.70 | 3.90 | 1.50 |
| 9 | Pure PPQ reference sample | 4.13 | 0.42 | 4.15 | 4.70 | 3.60 |

[a,b]—statistically significant differences were found for the mean values marked with different letters (ANOVA, $\alpha = 0.05$; Tukey's test, $\alpha = 0.05$).

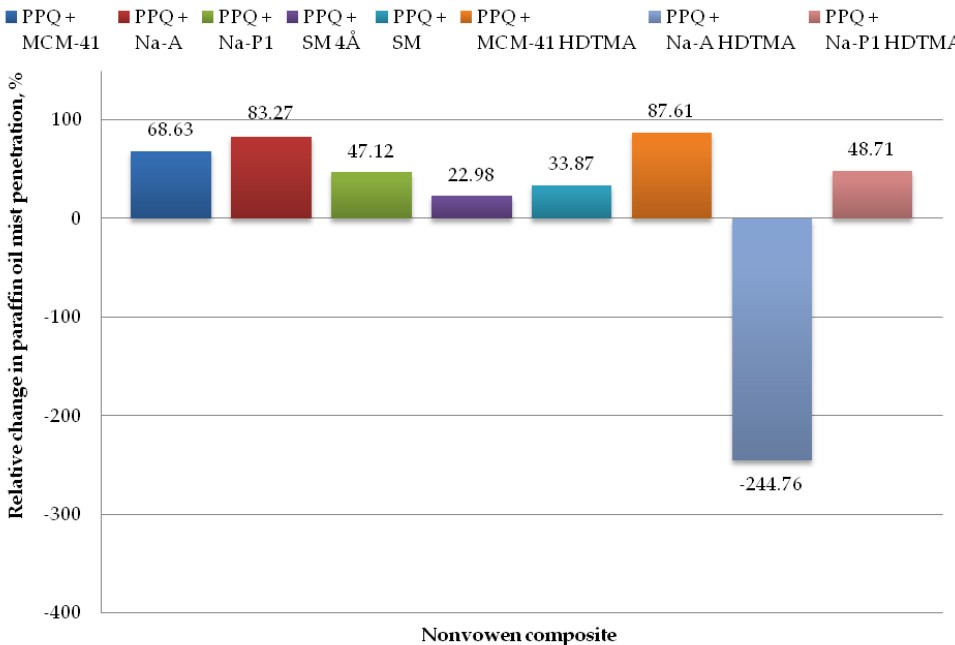

**Figure 11.** Relative change in paraffin oil mist aerosol penetration for nonwoven zeolite and silica composite.

The lowest paraffin oil mist penetration values were found for the composite with zeolite Na-A and unmodified and modified silica MCM-41 and MCM-41 HDTMA, at 0.69, 1.30, and

0.51%, respectively, with the corresponding relative changes amounting to 83.27, 68.63, and 87.61%, respectively (as compared to the reference sample without zeolite or silica materials). The introduction of mineral additives in the structure of elementary fibers in the melt-blown process improved filtration efficiency as compared to the reference nonwoven in three cases, that is, zeolite Na-A as well as unmodified and modified silica materials MCM-41 and MCM-41 HDTMA.

The highest paraffin oil mist penetration (14.25%) was found for the nonwoven composite containing the modified zeolite Na-A HDTMA, with the relative change being 244.76%. Similarly, as in the case of the NaCl aerosol, this is attributable to the fact that the applied zeolite contains chemical elements that may neutralize the electrostatic charges induced during the technological process.

### 3.5. Airflow Resistance

The mean airflow resistance values for the nonwoven composite materials containing zeolite and silica materials are given in Table 6 with the relative results shown in Figure 12. Deterioration of protective and functional properties of the composite materials in relation to the reference sample is marked with negative values, while the improvement of protective and functional with positive values. On the basis of the statistical analysis performed, significant differences for the average air flow resistance values are marked with different letters.

**Table 6.** Mean airflow resistance values for nonwoven composites containing zeolite and silica materials (descriptive statistics).

| No. | Variant | Mean Airflow Resistance, Pa | Standard Deviation | Median | MAX, Pa | MIN, Pa |
|-----|---------|------------------------------|---------------------|--------|---------|---------|
| 1 | PPQ + MCM-41 | 331.50 [b] | 14.38 | 334.50 | 348.00 | 313.00 |
| 2 | PPQ + Na-A | 648.00 [d] | 73.39 | 655.00 | 747.00 | 489.00 |
| 3 | PPQ + Na-P1 | 537.14 [c] | 40.37 | 520.00 | 590.00 | 485.00 |
| 4 | PPQ + SM 4Å | 268.00 [ab] | 4.69 | 266.50 | 275.00 | 261.00 |
| 5 | PPQ + SM | 283.50 [ab] | 34.32 | 278.50 | 338.00 | 250.00 |
| 6 | PPQ + MCM-41 HDTMA | 876.80 [e] | 36.42 | 851.00 | 940.00 | 848.00 |
| 7 | PPQ + Na-A HDTMA | 306.25 [ab] | 27.49 | 308.50 | 341.00 | 267.00 |
| 8 | PPQ + Na-P1 HDTMA | 517.40 [c] | 73.48 | 539.00 | 593.00 | 386.00 |
| 9 | Pure PPQ reference sample | 260.83 | 8.82 | 260.50 | 272.00 | 248.00 |

[a,b,c,d,e]—statistically significant differences were found for the mean values marked with different letters (ANOVA, $\alpha = 0.05$; Tukey's test, $\alpha = 0.05$).

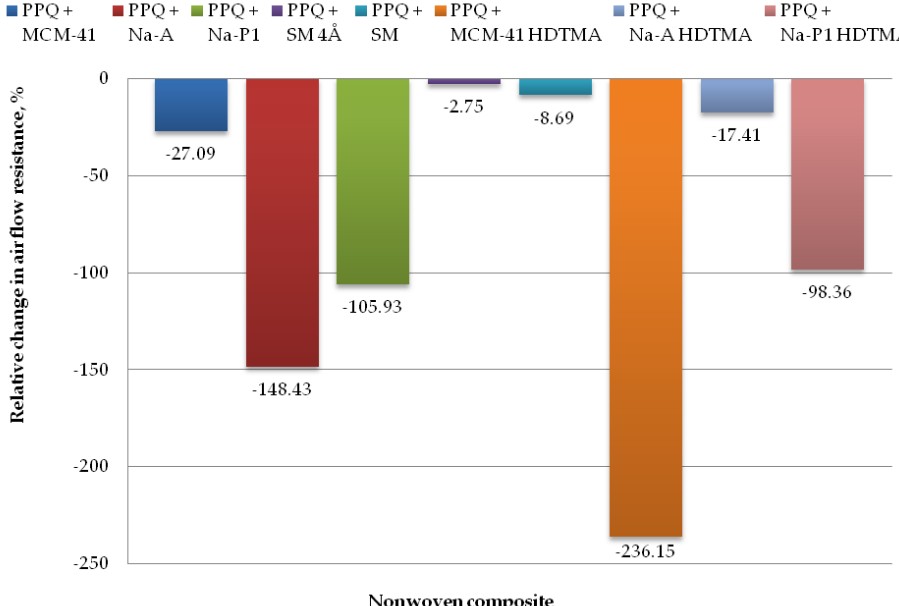

**Figure 12.** Relative change in airflow resistance for nonwoven zeolite and silica composite.

The lowest airflow resistance was found for the composites PPQ + Na-A HDTMA, PPQ + SM, and PPQ + SM 4Å, at 306.25 Pa, 283.50 Pa, and 268.00 Pa, respectively. The corresponding relative airflow resistance values were 17.41, 8.69, and 2.75%, respectively, as compared to the reference

sample. The introduction of additives during the nonwoven production process resulted in a slight increase in airflow resistance relative to the reference sample. In the case of the composite containing PPQ + MCM-41, airflow resistance increased by 27.1%.

The highest airflow resistance was found for the nonwoven composites containing zeolites Na-A and Na-P1, and modified silica MCM-41 HDTMA at 648.00 Pa, 537.14 Pa, and 876.80 Pa, respectively, with the corresponding relative changes of 148.43, 105.93, and 236.15%. The increased airflow resistance is associated with clogging by zeolite and silica particles, which filled the interfiber spaces. According to the EN 149:2001+A1:2009 clogging test, airflow resistance must not exceed 700 Pa at a flow rate of 95 L/min. The above criterion was not met for the composite containing the modified silica material MCM-41 HDTMA.

## 4. Conclusions

The presented morphological tests studies of zeolite and silica materials showed that the most widely used modification with HDTMA adversely affected the textual parameters of those materials. While it did not change their phase structure, it did clog their mesopores and decreased specific surface area. In addition, this modification was conducive to agglomeration during the introduction of zeolite and silica materials in the structure of elementary polymeric fibers in the melt-blown process.

The protection and functional tests of the studied nonwoven zeolite and silica composites in terms of breakthrough time revealed that the larger the BET specific surface area, the longer the reaction time with the vapors of volatile compounds (both organic and inorganic). The longest breakthrough time for ammonia vapors was observed for the composite containing the mesoporous silica material MCM-41, which was also characterized by high filtration performance against aerosols with both liquid and solid dispersed phases (NaCl and paraffin oil mist), ranging from 97% to 99%, as well as a satisfactory airflow resistance of approximately 330 Pa.

The application of the SM molecular sieve (zeolite ZSM-5) in the PP nonwoven led to the longest breakthrough time against acetone and cyclohexane vapors at the maximum allowable concentrations (235 and 81 ppm, respectively), with reaction times of approximately 11 and 2min, respectively. The presented basic studies of filtration performance showed the nonwoven composite with SM to be 97% effective against sodium chloride and paraffin oil mist aerosols at an airflow resistance of 283.5 Pa. The obtained results are in line with the results of research of other scientists [34–36], which showed that zeolites are good adsorbents for chemical compounds.

As a result of the tests carried out and on the basis of the obtained results, it turned out that the material MCM-41 shows the highest PC values for ammonia. This indicates a high potential of interaction between the adsorbent surface and the gas being tested, which confirmed the results of the research presented in the articles [37,38]. Similar observations were recorded for the zeolite SM material that showed the best efficacy against VOCs.

**Author Contributions:** Conceptualization, A.B.; Methodology, A.B.; Formal analysis, A.B., A.N., R.P. and W.F.; Investigation, A.B., A.N. and R.P.; Resources, A.B. and W.F.; Data curation, A.B., A.N., W.F. and R.P.; Writing—original draft preparation, A.B., A.N., W.F.; Writing—review and editing, A.B., A.N. and W.F.; Supervision, A.B. and W.F.

**Funding:** The publication is based on the results of Phase IV of the National Program "Improvement of Safety and Working Conditions" financed in the years 2017–2019 in the area of research and development by the Ministry of Science and Higher Education/National Centre for Research and Development. The Central Institute for Labour Protection—National Research Institute is theProgramme's main coordinator.

**Conflicts of Interest:** The authors declare no conflict of interest. The funders had no role in the design of the study; in the collection, analyses, or interpretation of data; in the writing of the manuscript, or in the decision to publish the results.

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
