# Peer review of "The Effects of Textural Parameters of Zeolite and Silica Materials on the Protective and Functional Properties of Polymeric Nonwoven Composites"

_applsci, doi:10.3390/app9030515_

Round 1

Reviewer 1 Report

 In table 4, why the unit of composite thickness is in 'g'?

The scale bar in SEM images are barely visible, and the images seem to be stretched or distorted?.

Why the relative change in NaCl is negative for PPQ and Na-AHDTMA?

Author Response

Dear Reviewer 1,

the responses to the review are included in the Word file.

Best regards,

Brochocka Agnieszka and co-authors.

Reviewer 2 Report

The topic debated in the manuscript is interesting and deserving a scientific study. The authors reports data concerning the sorption properties and penetration properties of several composites based on PP and zeolite or silica materials. However the paper is long, with a lot of data and rather confusing for the reader. Furthermore, despite in the abstract the authors say that “the objective was to analyze the influence of the textural parameters of zeolite and mesoporous silica materials on the protective and functional properties of polymeric nonwoven composites containing them” in my opinion the results are reported without an adequate discussion. Therefore I suggest the authors to improve in general the clarity of the manuscript and more specifically the discussion of the results in order to make well clear to the reader the relation among the parameters of the materials and their properties.  In addition there are some refuses or unclear points reported as follows:

-          There are two paragraphs 3.3

-          The paragraph 3.3. “Penetration by NaCl Aerosol” appears rather confusing and in particular the meaning of the last sentence in it (“This is attributable to the fact that the applied zeolite contains chemical elements that may neutralize the electrostatic charges induced during the technological process”) should be clarified

-          Paragraph “3.3. Penetration by Paraffin Oil Mist”: same observations as in the previous point

-          In the conclusion “mineralogical and structural study” is cited, but for me it is not clear what is the mineralogical study and also as structural study I have some doubt. Please clarify.

Author Response

Dear Reviewer 2,

the responses to the review are included in the Word file.

Best regards,

Brochocka Agnieszka and co-authors.

Reviewer 3 Report

Accept in the present form

Author Response

Dear Reviewer 3,

we hope that the Reviewer will be satisfied with our responses and revisions of the original manuscript. We also believe that the corrections made in the article will make it more valuable for readers of Applied Science.

Yours sincerely,

Agnieszka Brochocka and co-authors.

Reviewer 4 Report

ARGUMENT: The objective of the paper "The effects of textural parameters of zeolite and silica materials on the protective and functional properties of polymeric nonwoven composites" was to analyze the influence of the textural parameters of zeolite and mesoporous silica materials, modified with HDTMA or not, on the protective and functional properties of polymeric nonwoven composites containing them. Samples were characterized by BET and SEM and texted for sorption properties with ammonia, acetone and cyclopropane, penetration by sodium chloride and paraffin oil mist aerosols, airflow resistance.

NOVELTY: Given the scarcity of data in the literature on the application of zeolites and mesoporous materials as gas filters, for which mainly active carbons are used, the subject has a certain interest and novelty.

CONCLUSIONS: The results show some data about the positive performances in the application of ZSM-5 zeolite and MCM41 in PP nonwoven, but despite this, the authors' purpose of identifying the correlations of the material's structure and performance, fails. No clear correlation between structure and property has been identified or hypothesized, besides correlation between high surface area and amount of adsorbate particles (lines 268-270) that is well known and trivial. The role of HDTMA has not been clarified.

RESULTS: A very serious problem of this paper is that it shows in the tables a large amount of processing data that are not relevant for the paper and that have not even been discussed or commented, as well as unnecessary physi-chemical literature parameters.

Another problem with this paper is that some experimental details are lacking in sample preparation and characterization, and sample characterization is limited only to SEM images and BET analysis.

Before the eventual publication, it will be necessary a deep revision and implementation of the text with major revisions and with the elimination of many useless data basing on the following points:

Major revisions:

1) ”The studied composites were made with unmodified mesoporous silica materials and zeolites synthesized from fly ash by the classical hydrothermal method, as well as zeolite and silica materials modified with hexadecyltrimethylammonium (HDTMA) bromide supplied by the Lublin University of Technology (Table 1).”

Zeolites and mesoporous silica are all prepared by Lublin researchers? Or only the modified ones? It is not clear. How was the HDTMA addition made?  The preparation method s(defined classical) have to be explained more in detail.  

2) Which analysis confirmed the zeolite and silica structures? are they home-made? They have to be reported o added references.

3) Why did the authors study modified samples with HDTMA? What is the expected effect of adding HDTMA? Authors have to clearly explain your ideas and your purposes.

4) Table 1 is useless; it is a repetition of table 2. Table 1 must be eliminated. The column with the number is useless, which is never recalled in the text or in the following tables.

5) Table 3 contains only one sample. The reported literature parameters must be reported in the text and table 3 must be deleted.

6) only SEM characterization is widely discussed in the results. The results must be implemented with discussion of other performed characterizations.

7) The data in Table 4 have not been commented or discussed in the results. They miss details in the experimental: how surface density and composite thickness were determined?

8)”The concentrations of the test substances: acetone (235 ppm), cyclohexane (81 ppm), and ammonia (18.7 ppm) corresponded to their MAC levels pursuant to the Regulation of the Minister of Labor and Social Policy of June 12, 2018 [18]; the volumetric flow rate was 30 L/min.”

Regulation of which country? The authors must specify or change.

Authors should replace reference with regulation of international legislation.

9) Table 5 reports CAS number, Density and Molar Mass (!). They are chemical data or parameters well known and unless in the discussion. In addition Density is related to the liquid phase (?!), so not inherent at all in that research argument. Table 5 it does not make sense and have to be eliminated.

10) Line 203 “Approximately 55 g of zeolites Na-A and Na-A HDTMA 203 were incorporated per nonwoven sheet.”  line 214 “Approximately 123 g of zeolites Na-P1 and Na-214 P1 HDTMA were incorporated per nonwoven sheet.  Line 238 “(approx. 18 g per sheet),”

These information about the amount of zeolites:

i) have to be added in the table 4 for clarity

ii) why the amounts are so much different in the samples?

iii) how the figures may be compared in dispersion if  the added amount it is so different? Authors have to comment.

11) tables 6, 7 and 8: descriptive statistics:

the statistical data are not discussed in the text and are useless to the discussion, moreover it is not explained what exactly they refer to. They are absolutely incomprehensible and completely useless ad whit a-b subscripts not clearly explained. Only the "mean aerosol penetration" column may be of interest. The three columns "mean aerosol penetration", in the three corresponding tables 6, 7 and 8 must be reported in a single table, favoring the comparison between different vapours, whereas the descriptive statistics have to be eliminated. 

12) “The presented mineralogical and structural studies of zeolite and silica materials showed that the most widely used modification with HDTMA adversely affected the textual parameters of those materials”

SEM and BET are not a mineralogical study.

SEM and BET are not a structural study, but only a morphological study

Minor revisions:

 1) Line 118) and line 123)… the temperature is in °C and in Kelvin. The authors must standardize.

2) SEM results:  Figures 2-3-4-5-6-7-8-9: technical data of SEM acquisition in the black line at the bottom should be cut. Enlargement have to be specified in the caption.

3) Figure of unmodified and modified system (i.e. fig 2a-b and 3a-b) (fig 4 and 5), ( fig 6 and 7) have to be merged in one image, to help the readers and for a more clear exposition.

Author Response

Dear Reviewer 4,

responses to the Reviewer's review are included in the Word file.

Yours sincerely,

Agnieszka Brochocka and co-authors.

Reviewer 5 Report

The authors did an excellent job in using the micro and mesoporous materials such as zeolites and silica materials along with nonwoven composite to adsorb ammonia, cyclohexane and acetone as vapors in order to use in FFRs. They have utilized SEM and adsorption analyzers to characterize the materials for their properties. I would accept this paper after the authors can address these minor concerns given below.

1. On line 160, include space between “burner” and “as”

2. The authors explained that the PPQ+MCM-41 has highest adsorption capacity to ammonia, which is also seen in Fig 10. The authors attributed the highest capacity due to the high surface area of MCM-41. I didn’t understand why the same logic doesn’t apply to the adsorption capacity towards acetone and cyclohexane.

3. Also, I was wondering whether the authors tried to study stability of these modified and unmodified nonwoven zeolite and silica materials. Generally, zeolite materials are unstable to humid atmosphere. How does the zeolites respond to the adsorption of the malodorous vapors in the presence of water vapor?

Author Response

Dear Reviewer 5,

responses to the Reviewer's review are included in the Word file.

Yours sincerely,

Agnieszka Brochocka and co-authors.

Round 2

Reviewer 1 Report

Publish

Author Response

Dear Reviewer 1,

I would like to thank you in my and co-authours name for the opinion and involvement during the editing our manuscript.

Yours sincerely,

Agnieszka Brochocka and co-authors.

Reviewer 4 Report

The manuscript has been significantly improved and now it warrants the publication in Applied Sciences 

Author Response

Dear Reviewer 4,

I would like to thank you in my and co-authors name for the opinion and involvement during the editing our manuscript.

Yours sincerely,

Agnieszka Brochocka and co-authors.